# At a Crossroads to Cancer: How p53-Induced Cell Fate Decisions Secure Genome Integrity

**DOI:** 10.3390/ijms221910883

**Published:** 2021-10-08

**Authors:** Dario Rizzotto, Lukas Englmaier, Andreas Villunger

**Affiliations:** 1CeMM Research Center for Molecular Medicine of the Austrian Academy of Sciences, 1090 Vienna, Austria; DRizzotto@cemm.oeaw.ac.at (D.R.); Lukas.Englmaier@rud.lbg.ac.at (L.E.); 2Ludwig Boltzmann Institute for Rare and Undiagnosed Diseases (LBI-RUD), 1090 Vienna, Austria; 3Institute for Developmental Immunology, Biocenter, Medical University of Innsbruck, 6020 Innsbruck, Austria

**Keywords:** p53, cell cycle, DREAM-complex, cell death, PIDDosome, CIN, aneuploidy

## Abstract

P53 is known as the most critical tumor suppressor and is often referred to as the guardian of our genome. More than 40 years after its discovery, we are still struggling to understand all molecular details on how this transcription factor prevents oncogenesis or how to leverage current knowledge about its function to improve cancer treatment. Multiple cues, including DNA-damage or mitotic errors, can lead to the stabilization and nuclear translocation of p53, initiating the expression of multiple target genes. These transcriptional programs may be cell-type- and stimulus-specific, as is their outcome that ultimately imposes a barrier to cellular transformation. Cell cycle arrest and cell death are two well-studied consequences of p53 activation, but, while being considered critical, they do not fully explain the consequences of p53 loss-of-function phenotypes in cancer. Here, we discuss how mitotic errors alert the p53 network and give an overview of multiple ways that p53 can trigger cell death. We argue that a comparative analysis of different types of p53 responses, elicited by different triggers in a time-resolved manner in well-defined model systems, is critical to understand the cell-type-specific cell fate induced by p53 upon its activation in order to resolve the remaining mystery of its tumor-suppressive function.

## 1. Introduction

Despite decades of research, cancer remains one of the leading causes of death worldwide, and particularly in developed countries its incidence is still on the rise. Cancer usually arises upon long latency in response to a series of genetic alterations, triggered by exogenous (environmental exposition) and endogenous causes (genetic predisposition) that are additionally modulated by pathobionts, such as bacteria or viruses, making it very difficult to predict the time of its occurrence, its clinical progression, and treatability [1,2,3,4,5,6].

However, neoplastic cells originating from different tissues often display similar molecular features, for example, the overexpression or mutation of proto-oncogenes, such as *MYC* or *RAS*, or the inactivation of tumor-suppressor genes. Perhaps the most common gene of this kind that is inactivated or lost in human cancers is *TP53* (herein, p53). Nearly all the events currently known to compromise the genomic integrity of the cell (thus being potential drivers of transformation) can lead to p53 activation. This requires protein stabilization, accumulation, and translocation into the nucleus, where p53 can elicit its function as a sequence-specific transcription factor [7]. Moreover, direct cell-death-promoting effector functions have also been proposed and will be discussed later.

Stunningly, despite extensive research into its biology and more than 100,000 papers cited in PubMed, we still fall short in understanding how p53 actually prevents cancer and how we could exploit current knowledge therapeutically.

The two best-studied cellular responses controlled by p53 are cell cycle arrest and the induction of apoptosis, both of which aim to preserve genomic integrity for tissue homeostasis [8,9,10,11]. Indeed, active p53 promotes the transcription of *CDKN1A*, the gene encoding the potent cyclin-dependent kinase inhibitor p21 [12], and several genes encoding proteins involved in regulating mitochondrial and death-receptor-driven apoptosis [8,13]. Yet, cell cycle arrest and apoptosis are only two of the many tumor-suppressive functions controlled by p53 [14,15,16]. Consistently, impairing the ability of p53 to activate *CDKN1A* and its apoptotic effectors does not impair its ability to suppress cancer [17,18]. This suggests additional mechanisms at play, likely those reducing the mutational burden and the maintenance of genome integrity [8,19,20]. Yet, alteration or loss of either of these effector’s arms can contribute to tumorigenesis as well as drug resistance phenotypes, as documented in multiple studies [21,22,23,24,25,26].

The importance of p53 in maintaining genome integrity becomes clear when looking at malignant cells in which the transcription factor is lost, mutated, or inactivated. Indeed, a common feature of these cells is genomic instability, which is reflected by the increased accumulation of alterations at different levels, from mutations of the genetic sequence to alterations in chromosomes’ structure and number [27]. Alterations noted at the level of the chromosomes are summarized as chromosomal instability (CIN) or, upon fixation, aneuploidy. CIN arises from errors during the mitotic process, caused by defects in checkpoints controlling mitotic entry, regulated by checkpoint kinase 1 (CHK1) [28], or mitotic exit, controlled by the E3-ligase, APC/C, and its activator CDC20, as well as the spindle assembly checkpoint (SAC), ensuring the integrity of kinetochore-microtubule attachments in prometaphase [29,30]. Moreover, errors in centrosome biogenesis and number can also foster CIN, e.g., during multipolar mitoses [31,32,33,34]. Impairment of any of these control mechanisms can cause the missegregation of chromosomes into daughter cells, resulting in aneuploidy, an ultimately fixed abnormal number of chromosomes deviating from the physiological karyotype [35]. CIN and aneuploidy are interconnected [27], and they contribute to increasing intra-cancer heterogeneity, which in turn can confer a selective advantage, for example, in the development of resistance to conventional or targeted therapies [36,37]. Nonetheless, aneuploid cells often show initially decreased fitness, proteostasis, and proliferation defects, together with an increased susceptibility towards cell death [38,39,40]. This suggests that aneuploidy might be only beneficial in certain contexts after sampling the right set of chromosomes, allowing survival and outgrowth of cells with complex karyotypes. Hence, aneuploidy is often considered a rather late event in tumorigenesis, driving cancer evolution, but may not necessarily be responsible for tumor initiation [31,36]. Yet, this is an ongoing debate [38].

As stated before, p53 inactivation is a common feature of many cancers. The most prominent way of inactivating p53 is through mutations, often occurring in the DNA binding domain of the transcription factor [41,42,43]. Germline mutations in the *TP53* gene are associated with the Li-Fraumeni syndrome, whose patients are prone to develop multiorgan cancers, often in the childhood [44,45]. Nevertheless, inherited single nucleotide polymorphisms (SNPs) have been identified in the *TP53* loci and have been reviewed and discussed in detail elsewhere [46,47]. The impact of SNPs on tumor susceptibility is not easy to assess mechanistically (using both in vitro and animal models). To exemplify this complexity, one of the most common polymorphisms affecting p53 occurs on codon 72, which encodes for either a proline (P72) or an arginine (R72). The R72 variant has been shown to correlate with the establishment of a successful apoptotic program upon p53 activation in cell models [48], also due to a better induction of *CDKN1A* and *BAX* [49]. Nevertheless, in a breast cancer mouse model, the P72 variant seems to be protective against carcinogenesis compared to the R72 variant [50]. Cohort studies in cancer patients and meta-analyses did not lead to univocal results, such as, for example, the R72 variant, which seems to favor HPV-associated cervical carcinoma in vitro [51] but does not lead to an increased risk of cervical cancer in patients [52]. Yet, this variant has been identified as a risk factor for breast cancer [53,54]. Further analyses aimed at assessing the impact of the different SNPs are needed to clarify their contribution to tumorigenesis.

Interestingly, genetic variants affecting genes of the p53 pathway (both upstream activators of the transcription factor and the downstream targets) influence cancer susceptibility [55].

## 2. How p53 Puts the Break on CIN and Aneuploidy

CIN and aneuploidy are frequently preceded by defects in cytokinesis, referring to the process allowing the mother cell that has duplicated its genome to physically separate into the two daughter cells at the end of mitosis [56,57]. In this situation, the cell undergoes whole-genome duplication, most commonly leading to tetraploidization [57]. It is believed that tetraploidization is an early event occurring in many cancers [58], which can foster CIN and aneuploidy in subsequent rounds of cell division. Moreover, tetraploid cells are more tolerant to chromosome loss compared to their diploid counterparts and consequently are more “fit” to undergo cancer-promoting alterations [36]. Another consequence of cytokinesis failure is the acquisition of extra centrosomes [59], which are microtubule-organizing centers (MTOCs) responsible for generating the mitotic spindle during prophase [60]. In mitosis, the presence of extra centrosomes can lead to multipolar cell division, which is highly error-prone [9,59] due to frequent merotelic kinetochore–microtubule attachments [59]. Not surprisingly, extra centrosomes are frequently seen in cancer lesions and are also discussed to increase the invasiveness of tumor cells [61,62].

Remarkably, p53 signaling can halt cells harboring an altered centrosome number as well as cells experiencing problems during mitosis that primes them for chromosome mis-segregation [63]. As an example, extended mitotic duration due to persistent activation of the SAC, which senses unattached kinetochores not bound to the mitotic spindle, can trigger a p53-dependent cell-cycle arrest [64,65,66] (Figure 1). This mechanism relies on 53BP1 and USP28 that stabilize p53 by removing ubiquitination performed by MDM2, the main E3-ligase controlling p53 protein levels. This allows its accumulation and the induction of p21, arresting the cell in the next G1 [64,65,66]. Of note, the quality of this type of p53 response is expected to differ from the one induced upon DNA damage, where 53BP1 is known for rapidly accumulating on chromatin at the sites of double-strand breaks and providing a scaffold for factors involved in DNA repair [67]. Under these conditions, p53 stabilization primarily depends on post-translational modifications (PTMs) mediated by kinases of the DNA damage response (DDR) pathway, such as ATM, ATR, CHK1, and CHK2, that phosphorylate p53 on specific N-terminal amino acid residues (e.g., Ser15, Ser20, and Ser37) [68,69], leading to the displacement of MDM2 for p53 stabilization (Figure 1). In the context of DNA damage, 53BP1 has a crucial role in directing non-homologous end-joining or homologous recombination to remove double-strand breaks [67,70]. Despite its name (p53 Binding Protein 1) and the presence of a domain at the c-terminus that can actually interact with p53, the importance of the interaction between p53 and 53BP1 remained uncertain [67]. More recently, it has been demonstrated that 53BP1, in concert with USP28, directly tunes p53 transcriptional activity in response to DNA damage independently of its function in promoting DNA repair [71]. Nevertheless, 53BP1 KO, as well as USP28 KO cells, retain p53 stabilization in response to the DNA-damage-inducing agent doxorubicin, while phosphorylation of p53 on Ser15 does not occur in cells experiencing extended mitotic duration [65]. Overall, this suggests that, despite the involvement of the same actors, the downstream effects imposed by p53 in order to control cell fate could diverge in response to DNA damage and extended mitotic duration, given the different mechanisms and interactors that lead to p53 stabilization and activation.

Maybe unsurprisingly, the accumulation of extra centrosomes, e.g., as experienced by cells after defective cytokinesis, utilizes yet again a different machinery to activate p53, limiting their expansion or survival [72]. Here, a multiprotein complex, dubbed the PIDDosome [73], is engaged to promote the activation of a cysteine-driven protease, caspase-2, that can cleave MDM2, to neutralize this E3-ligase and enable p53 stabilization. Of note, the N-terminal fragment of MDM2, devoid of its E3-ligase domain, remains attached to p53, and phosphorylation events noted during the DDR are not seen upon centrosome accumulation [72], again pointing towards qualitative and potentially quantitative differences in this type of p53 response (Figure 1).

## 3. P53-Induced Cell Cycle Arrest and Senescence

As stated previously, the induction of cell cycle arrest appears intuitively helpful to prevent the outgrowth of a cell that has lost genome integrity, but this response, even in its most stringent form (i.e., senescence) appears to be overcome eventually during transformation or tumor therapy.

The potent cyclin-dependent kinase inhibitor *p21* (*CDKN1A*) was the first transcriptional target of p53 to be identified as capable of regulating tumor growth upon p53 activation [12]. As other genes connected to the control of cell cycle, the *CDKN1A* locus contains two strong p53 response elements (at −2.3 and −1.4 kb from the transcription start site [74]), which allow a quick transcriptional upregulation of the *CDKN1A* gene upon p53 activation. The immediate result of p21 expression is the arrest of the cell cycle, which occurs via a p21-mediated inhibition of the cyclin/CDK complexes by physical interaction [21]. Inactivation of the cyclinE/A-CDK2 and cyclinD-CDK4/6 complexes prevents CDK-mediated phosphorylation of pRB, preventing the release of E2F transcription factors controlling the transition from G1 to the S phase [75,76] (Figure 2). Yet, by looking at the p53 core transcriptional program, additional genes besides *CDKN1A* appear to be related to cell cycle control [77]. Of note, GADD45A is known to act in concert with p21 and SFN (also known as 14-3-3-σ, another p53 transcriptional target) to inhibit the cyclin B1/CDK1 complex necessary for entry into mitosis [78,79]. In addition, p53 induces PLK2 and PLK3 that encode kinases belonging to the polo-like family, both of which play important roles in the maintenance of genome integrity in response to mitotic errors and DNA damage [80] (Figure 2). Moreover, a number of p53 targets that may indirectly impinge on cell cycle control have been reported, including *DUSP14, CyclinG1, BTG2, NUPR1, ZMAT3,* and *ZNF385A.*

Besides transcriptional upregulation, high-throughput studies on cell lines exposed to p53 activating treatments repeatedly identified a plethora of genes whose expression was downregulated. Initially, it was considered that p53 could directly dampen the expression of these genes; therefore, different models that could explain the repressive mechanism have been proposed. Nevertheless, experimental evidence from ChIP data failed to determine a direct connection between p53 binding to a DNA sequence on a given gene and a subsequent reduction in mRNA production [7].

## 4. P53 and the DREAM Complex

The discovery of the DREAM complex was crucial to disentangle the mechanisms of p53-mediated target gene repression, which are now considered to be exclusively indirect through DREAM [77,81,82] and strictly dependent on the presence of p21 [83]. The DREAM complex (dimerization partner (DM), RB-like, E2F, and multi-vulval class B (MuvB)) is multiprotein machinery that, depending on the subunits interacting with the MuvB core, can bind with different regulatory elements on the promoter of target genes and, importantly, can act both as a transcriptional activator or repressor in a context-dependent manner [84]. A detailed review of the DREAM complex can be found in [85]. In physiological settings, cells are held in G1 (or G0) by the MuvB interacting with the pocket proteins p107 and p130, both of which are structurally and functionally related to pRB that bind, in turn, the repressor E2F4/5 (DREAM complex), resulting in downregulation of pro-proliferative E2F-responsive genes (Figure 2). Upon phosphorylation of p107/p130 by the cyclin/CDK complexes, the interaction with the MuvB core is terminated, allowing MuvB to bind B-MYB and/or FOXM1, forming the MMB (bMyb-MuvB) or the MMB-FOXM1 complexes. The latter complexes promote the transcription of target genes responsible for cell cycle progression through S, G2, and M phases [86,87]. Conversely, following p53 activation, p21-dependent inactivation of cyclin/CDKs complexes prevents the phosphorylation of p107/p130, arresting the switch from DREAM to MMB/MMB-FOXM1 complexes and repressing the transcription of target genes [88]. In this way, p53 indirectly downregulates the expression of several genes involved in DNA replication, the G2 checkpoint, and mitosis, expanding its cell cycle regulating potential beyond the simple induction of GADD45A, SNF, and PLK2.

P53-induced senescence is considered a stable form of cycle arrest that involves alterations in cell metabolism, gene expression, and chromatin composition and association with a specific senescence-associated secretory phenotype (defined SASP) [89]. Induction of senescence is considered a multistep process, in which the initial cell cycle arrest is exerted by p21, becoming permanent via the subsequent activation of the CDK inhibitor p16/INK4a [89]. Intriguingly, p53-induced senescence could also be reverted upon p53 inactivation and, more effectively, upon pRB inactivation [90,91,92]. Overall, cellular senescence seems to be a dynamic phenotype, whether it is p53-dependent or not [93]. Similar to transient cell cycle arrest, the DREAM complex contributes to p53-induced senescence. It was initially discovered that RNAi-mediated downregulation of the DREAM complex proteins LIN9, LIN54, and B-MYB results in premature senescence through activation of p53 in human fibroblasts. Moreover, interfering with the phosphorylation of LIN52 (a protein composing the MuvB core) by downregulation of the kinase DYRK1A prevents the switch of the MuvB core from activating to the cell cycle-repressive DREAM state, impinging the cells’ ability to enter senescence [94].

Of note, knock-down of *LIN9* in human fibroblasts in combination with p53 inactivation via SV40 results in high levels of aneuploidy [95]. A similar result was also observed in *LIN9*-deficient mouse embryonic fibroblasts (MEFs) that displayed a high degree of polyploidization, binucleation, and other nuclear abnormalities 24 h after cell cycle re-entry, which ultimately led to premature senescence [96]. These aberrations arise from defective cell division, such as prolonged mitotic timing, failed cytokinesis, acquisition of extra centrosomes, and formation of multipolar spindles in subsequent rounds of cell division. Not surprisingly, impairing the DREAM complex formation interferes with the cell division process, as the expression of many genes responsible for mitosis are controlled by the MuvB core in the repressor (DREAM) or activator (B-MYB/FOXM1) conformation. However, the high degree of aneuploidy reached by *LIN9*-deficient MEFs raises the question whether the activation of p53 and p21 (PIDDosome- and/or USP28-53BP1-dependent) is sufficient for preventing tumorigenesis and to what extent the downstream activation of p21 and DREAM is essential to prevent the outgrowth of multinucleated and/or aneuploid cells.

Recently, an interesting feedback loop was discovered that also connects the PIDDosome to the p53-p21 axis and cell cycle control. Indeed, besides being regulated by p53 (as PIDD1 is a transcriptional target of p53 itself [97]), Sladky et al. discovered that during normal liver development or during regeneration, transcription factors of the E2F family control the expression of *PIDD1* and *CASP2*. In this context, E2F1 promotes cell cycle progression as well as the upregulation of *CASP2* and *PIDD1*, which limit the polyploidization of liver cells. Conversely, E2F7 and E2F8 have an opposing role, reducing the expression of the PIDDosome components and allowing hepatocytes to reach their physiological polyploid status [98].

## 5. P53-Induced Apoptosis

Apoptosis is the best-characterized form of p53-induced cell death (reviewed in [8]). Several transcriptional targets of p53 are tied to the intrinsic mitochondrial or extrinsic death receptor (DR)-driven apoptotic pathways [77,99]. While differing in trigger and execution, both pathways depend on cysteine-aspartic proteases termed caspases to coordinate non-immunogenic apoptosis.

Certain cytotoxic insults such as DNA damage caused by UV or ionizing radiation can activate the intrinsic apoptotic pathway through the p53-driven expression of pro-apoptotic members of the BCL2 protein family [25,100,101,102,103]. Being comprised of pro-survival- and pro-apoptotic proteins, the BCL2 family members keep each other in check through direct protein–protein interactions mediated through their shared BH (BCL-2 homology) domains (reviewed in [104]). Pro-survival members, such as BCL2, BCLX, and MCL1, bind and sequester pro-apoptotic “BH3-only” proteins, including BID, PUMA, and NOXA, impairing their ability to activate the apoptosis effector proteins BAX and BAK [105] (Figure 3). Amongst these apoptosis regulators, well-characterized p53 transcriptional targets include the BH3-only proteins *PUMA/BBC3* [100], *NOXA/PMAIP1* [103] and potentially *BID* [106], as well as the apoptosis effector *BAX* [107,108]. These findings imply that p53 activity has direct consequences on apoptosis induction as its transcription of pro-apoptotic genes contributes to surpassing the apoptotic threshold.

Additionally, p53 may trigger apoptosis at mitochondria independently of transcription through direct protein–protein interactions. These include the direct engagement of mitochondria and binding of the pro-survival proteins BCL2 and BCLX [109,110]. Cytosolic p53 was further reported to directly activate BAX [111] and BAK upon mitochondrial translocation [112]. Mechanistically, p53′s cytosolic pro-apoptotic function upon genotoxic stress has been suggested to be regulated through the transcription of its target gene *PUMA*, which in turn disrupts cytosolic BCLX/p53 complexes by competitive binding [111,113]. p53 binding of BCLX is meanwhile supported by crystal structure data [114] and has been successfully targeted in glioblastoma xenograft models to induce apoptosis [115]. Its interaction with the apoptosis effectors BAX and BAK, however, has been mainly addressed with recombinant protein studies or in overexpression settings. While these findings are intriguing, they remain controversial, as physiological settings where p53′s direct protein–protein interactions become rate-limiting for intrinsic apoptosis remain to be uncovered.

Once activated, BAX and BAK homo-oligomerize in the outer mitochondrial membrane, leading to the release of apoptotic mediators, including cytochrome c and caspase-derepressing proteins. In the cytoplasm, cytochrome c interacts with APAF1 to promote the formation of the apoptosome, the activation platform of initiator caspase-9. Active caspase-9 can then proteolytically activate executioner caspases (caspase-3, -6, -7), which orchestrate the breakdown of the cell. Intriguingly, p53 seems to tune apoptosis by regulating *APAF1* expression [116,117]. Conversely, the p53-induced apoptosis inhibitor TRIAP1 has been reported to interfere with apoptosome formation [118,119] (Figure 3).

Extrinsic apoptosis is elicited by DRs that are transmembrane-signaling molecules belonging to the tumor necrosis factor receptor (TNFR) superfamily (reviewed in [120]). P53 also intersects this pathway as it can regulate the expression of several DRs. This includes Fas (CD95/APO-1) [121] and the TRAIL receptors 1 (DR4/TNFRSF10A) [122] and 2 (DR5/KILLER/TNFRSF10B) [123] that, respectively, bind the apoptosis-initiating ligands FasL and TRAIL (TNF-related apoptosis-inducing ligand). Ligand binding clusters these receptors, thereby forming signaling scaffolds via their specialized intracellular death domains (DD). These domains allow the recruitment of adaptor proteins such as FADD (Fas-associated death domain), forming a death-inducing signaling complex (DISC), which serves as an activation platform for the initiator caspases 8 and 10 [124,125,126]. In a cell-type-specific manner, these initiators can directly cleave and activate effector caspases (e.g., in lymphocytes) or drive apoptosis via the engagement of the mitochondrial apoptotic pathway (e.g., in hepatocytes). The latter is mediated by the BH3-only protein BID, a reported p53 target, which is truncated and activated by caspase-8 [127,128] (Figure 3). On the contrary, through its transcriptional control of the DD lacking decoy receptors TRAIL receptor 3 (TRID/TNFRSF10C) and 4 (TRUNDD/TNFRSF10D), p53 has also been reported to negatively regulate apoptosis [129,130].

Albeit the transcriptional activation of DR family proteins by p53 remains undisputed, its role in apoptosis in response to genotoxic stress in cancer versus normal cells remains to be fully understood. It remains plausible that the expression of these TNFR family proteins by p53 contributes to sterile inflammatory signaling after cells experience DNA damage or undergo mitotic errors.

## 6. Non-Apoptotic Cell Death Forms Regulated by p53

Entosis is a form of non-apoptotic cell death in which one cell invades another, forming a cell-in-cell (CiC) structure. The presence of these structures has been noted in histological examination of tumor specimens at the end of the nineteenth century. It can involve different types of cells (both in terms of tissue derivation and oncogenic status) but not professional phagocytes [131,132]. Mechanistically, entosis requires actomyosin rearrangements that are driven by the GTPase RhoA and the effectors ROCK1 and ROCK2. Cytoskeletal rearrangements on the internalizing cell push it towards the future host, and the invasion process requires adherent junction formation between the entotic and the host cell [133]. Once internalized, most entotic cells undergo cell death by a mechanism involving their lysosomal degradation. However, sometimes, cells are either released from or remain alive within the host for extended periods of time [133]. This can interfere with the host cell division cycle and ultimately lead to aneuploidy [134]. Consistently, entosis can be either tumor-suppressive or -promoting, as the internalized cell can undergo cell death but also interfere with normal cell division of the host, promoting aneuploidy. More recent reports identified how the p53 status of the invaded host can discriminate the pro- or anti-tumorigenic aspect of entosis. Mackay and colleagues [135] identified how a host cell bearing mutant p53 is more prone to form CiC structures and take advantage of the engulfed cell to create genetic diversity via aneuploidy, while Durgan et al. determined how adherent epithelial cells undergoing mitotic defects (such as prolonged metaphase arrest and/or changes in mitotic morphology) are more prone to being internalized by neighboring healthy cells. This phenomenon can potentially prevent the outgrowth of cells bearing genomic defects [136] (Figure 3). Along this line, Liang et al. determined how entosis following mitosis is controlled by p53, which is activated upon the DNA damage experienced by the cell undergoing prolonged mitotic arrest. In this context, p53 directly upregulates the expression of its target *Rnd3*, which directs the function of the RhoA-ROCK1 pathway to remodel the actomyosin filaments promoting the penetration of neighboring cells [137]. In this case, prolonged mitosis and the activation of the 53BP1-USP28 axis is not responsible for entosis, suggesting that p53 controls two independent mechanisms aimed at preventing the growth of cells experiencing difficulties in metaphase: on the one hand, entosis and clearance by its neighbors, and on the other hand, the p21-mediated arrest in the next interphase [64,65,66]. If the 53BP1-USP28 axis can trigger p53-induced apoptosis in cells entering the G1 phase remains to be investigated.

Ferroptosis is a non-apoptotic, caspase-independent form of cell death caused by extensive iron-dependent lipid peroxidation [138]. By altering the intracellular redox balance, p53 can both induce and block ferroptosis (reviewed in [139]).Physiological amounts of oxygenated phospholipids in the plasma membrane are reduced by the glutathione peroxidase GPX4 to prevent ferroptosis [140]. Through transcriptionally repressing SLC7A11, p53 reduces available intracellular antioxidant levels and promotes oxidative damage, leading to ferroptosis induction [141]. SLC7A11 is part of the cystine/glutamate antiporter system x_c_^-^ that is essential for sustaining reduced glutathione pools. In vivo studies found that mice harboring an acetylation-deficient, mutated form of p53 (3KR), which renders them resistant to apoptosis, cell cycle arrest, and senescence, were still able to repress SLC7A11 and promote ferroptosis induction [141]. Moreover, mice expressing this 3KR p53 mutant and lacking the DNA damage repair gene XRCC4 were found to be viable (while XRCC4-deficient mice are not) and protected from spontaneous tumorigenesis, although showing high degrees of genomic instability [142]. As the authors found SLC7A11 downregulation and ferroptosis induction in these mice, they linked the regulation of ferroptosis to p53′s function as a tumor suppressor (Figure 3). Another pro-ferroptotic gene controlled by p53 is the polyamine-metabolizing enzyme SAT1 [143]. SAT1 overexpression sensitizes cells to ferroptosis, possibly through its downstream effects on the lipoxygenase ALOX15 [143]. When overexpressed in tumor cells in xenograft models, SAT1 limits tumor growth through ferroptosis induction, which was later shown to be additionally dependent on the lipoxygenase ALOX12 [143,144]. These findings provide further evidence on ferroptosis as a tumor-suppressive effector arm of p53. Additional target genes involved in controlling this type of cell death include FDXR (ferredoxin reductase) [145] and GLS2 (glutaminase 2) [146]. Yet, as mentioned above, p53 serves antithetical functions in ferroptosis regulation, which may be explained by cell-type-specific pathway alterations or PTMs such as acetylation [147]. For example, in wildtype colorectal cancer cells, p53 was found to suppress erastin-induced ferroptosis in a transcription-independent manner. This was suggested to rely on sequestration of dipeptidyl peptidase 4 (DPP4) in the nucleus, thereby limiting membrane peroxidation [148]. Another recent study found that the long-term stabilization of p53 and transcription of the cell cycle inhibitor p21 markedly delayed ferroptosis onset upon blocking of system x_c_^-^ [149]. This delay required available, reduced glutathione pools, suggesting that cell cycle stage and metabolic activity timed by p53 affects a cell’s susceptibility to undergoing ferroptosis [149]. Clearly, cell type and metabolic state will affect the outcome of p53-dependent changes in the signaling networks modulating ferroptosis susceptibility. Additional studies are needed to put these observations into context.

Mitosis is a tightly regulated process, as karyotype aberrations resulting from abnormal cell division can increase the susceptibility of daughter cells to malignant transformation (reviewed in [27]). To maintain genome integrity, the outgrowth of aneuploid and chromosomally instable cells is attenuated by processes summarized as mitotic catastrophe (reviewed in [150]). Caspase-2 has been promoted as a key executor in these pathways, which at least in part depend on p53 [151]. Caspase-2 has been implicated in a variety of biological functions, including apoptosis induced by DNA damage (reviewed in [152]). In this context, Tinel and Tschopp identified the PIDDosome as an activation platform of caspase-2 [73]. P53 activation has been seen as a crucial determinant of caspase-2-induced cell death, primarily through the transcriptional regulation of *PIDD1* [97]. While mouse genetics did not support a decisive role for the PIDDosome in DNA damage [153,154], in vitro studies provided evidence that caspase-2 can cleave MDM2, the master regulator of p53 protein levels, upon DNA damage, placing p53 simultaneously up and downstream of caspase-2 [155]. How can this conundrum be reconciled? As it seems, both scenarios can be true. The PIDDosome/MDM2/p53 axis is central in the response to centrosome accumulation, e.g., after cytokinesis failure [72], placing caspase-2 upstream of p53. Alternatively, PIDDosome-activated caspase-2 has been proposed to act as a fail-safe mechanism by initiating a stable p53 response in cells that escape cell cycle arrest following DNA damage [156]. Here, caspase-2 may act as an important safeguard of genome integrity downstream of p53. Consistently, γ-irradiated caspase-2-null MEFs were found to fail to undergo apoptosis and continue cycling [157]. Furthermore, MEFs, as well as MYC-induced B cell lymphomas from mice lacking caspase-2, show defective p53 signaling and increased aneuploidy [158,159]. Yet, the role of caspase-2 in clearing aneuploid cells as well as potential mechanisms involved needs to be clarified in future studies.

The list of “non-canonical” forms of cell death with reported p53 involvement is long and extends beyond the ones discussed above, including also autophagic cell death (ACD) and paraptosis (reviewed in [160]). Unsurprisingly, ACD can be limited by the downregulation of autophagic genes [161]. One important modulator worth mentioning is the p53-induced gene *DRAM1* that encodes a lysosomal sensor protein [162,163]. This transcriptional p53/DRAM-axis contributes to death induction upon ionizing radiation treatment in model cell lines [164]. Further studies report p53 phosphorylation and activation by MAP kinases following DR or TNFR stimulation, leading to the upregulation of autophagic executors, including DRAM or Beclin1 [165,166]. Consistently in both studies, p53 deficiency or inactivation rendered cells less susceptible to ACD. This will allow the correct placement of p53 in these pathways, considering its cross-talk with autophagy-defining mTOR signaling [167] and p53′s regulation at the protein level by the core autophagic complex Beclin1/Vps34 [168].

Paraptosis is another form of programmed cell death that is insensitive to caspase inhibitors and morphologically defined by the swelling and vacuolization of the endoplasmic reticulum and mitochondria [169]. A link between p53 and paraptosis has been suggested as drug-induced vacuole formation decreased in p53 knockout cells [170]. In vivo studies supporting a role of paraptosis or ACD in disease are rare, but in a mouse model investigating Alzheimer disease phenotypes, mice expressing a transgene of the p53 isoform p44 and the amyloid precursor protein were found to suffer from neurodegeneration that was caspase- and apoptosis-independent and shared morphological features assigned to paraptosis and ACD [171]. Yet, paraptosis can be effectively induced in several p53-defective cancer cell lines [172], highlighting the need for further studies for clarification. While the diverse functional portfolio of p53 links it to many cell death pathways, it can be assumed that p53′s function in some is more that of an indirect facilitator or amplifier.

## 7. Life–Death Decisions by p53—Flicking the Switch

What ultimately defines cell fate after p53 activation in diverse cell types? Over the years, different models have been proposed trying to explain how p53 imposes different phenotypes upon its activation, culminating in the re-entry into the cell cycle, the maintenance of an arrested phenotype (senescence), or the induction of cell death. As it seems, a consistent and universal predictor of the terminal phenotype is still to be determined or simply does not exist. Intuitively, the type of damage experienced by the cell, its severity, and duration in time can be crucial factors leading to either survival or death [16,173]. Indeed, an acute or sustained stressor will promote a higher magnitude of p53 activation or a sustained activation in time, which can result in the binding of weaker response elements, such as those situated on pro-apoptotic genes [9]. Nevertheless, this model does not hold universally true, e.g., cells belonging to different tissues show a different sensitivity towards the same p53-activating trigger [174,175], hinting towards a more complex network regulating the choice between life and death.

Besides, p53 is a protein that is subject to extensive post-translational modifications, which are often connected to the activating route of p53 itself and influence its functions [176,177]. Nevertheless, to the same subset of post-translational modifications, a completely different phenotype can follow [178] or, conversely, different routes of activation (leading to a different subset of PTMs on p53) can converge into the same terminal phenotype. An example is the induction of apoptosis correlated to extensive DNA damage, which is associated with phosphorylation on Ser46 [179,180]. Nevertheless, Nultin-3 treatment, which does not induce Ser46 phosphorylation [181], is still able to trigger p53-dependent apoptosis [175], suggesting additional layers of decision making.

The discovery of the pulsatile nature of p53 activity over time [182] opens a different scenario in the determination of the choice between life and death, as cells showing an oscillating activation of p53 are more prone to repair DNA damage and survive, whereas a sustained p53 activation (either cell-type-specific or treatment-dependent) is associated with senescence [183] and ferroptosis resistance [149]. The oscillatory behavior observed is due to the feedback loop between p53 and MDM2 [182]. If cells slip out of the arrest and re-enter the cell cycle before the DNA damage is completely repaired, p53 is re-activated with sustained dynamics thanks to the action of the PIDDosome, which inactivates MDM2 upon completion of mitosis in the presence of unrepaired DNA damage and allows the establishment of sustained p53 activation [156]. Conversely, apoptosis is only induced upon a quicker induction of the protein compared to surviving cells and is triggered only in cells that reach a certain threshold of p53 accumulation [184,185]. Surprisingly though, activation dynamics do not influence the binding potential of the protein to its response elements [186], suggesting that phenotype decision occurs post-transcriptionally. This is in line with the findings of Andrysik and collaborators [77]: cell lines exposed to the same p53-activating stimulus but showing a different outcome to the treatment (i.e., cell death or apoptosis) share a nearly identical transcriptome (also comprising apoptotic modulators in cell-cycle-arresting cells). However, at the translatome level, the ensemble of mRNA that is actively translated changes dramatically between cell lines, where apoptosis mediators are bound to ribosomes and translated much before apoptosis onset in sensitive cells [77]. The translatome of apoptotic cells is strongly influenced by the intrinsic repertoire of RNA-binding proteins (RBPs) that are present in those cells, shaping the phenotype at a post-transcriptional level [187]. Intriguingly, p53 itself can directly participate to post-transcriptional gene regulation by controlling the transcription of many RBPs and microRNAs [188,189].

## 8. Conclusions

p53 is at the center of a branching network of effector programs, including cell cycle arrest, senescence, and various forms of cell death. How exactly p53 makes the decision of which program to initiate remains elusive but is at least in part stimulus- and cell type-dependent. As the complexity of the p53 responses observed and the network of players is constantly growing and is far from being disentangled, research on the most studied tumor-suppressor gene needs to continue to provide satisfactory context- and cell-type-dependent answers that will eventually help to harness p53 for cancer treatment.

## Figures and Tables

**Figure 1 ijms-22-10883-f001:**
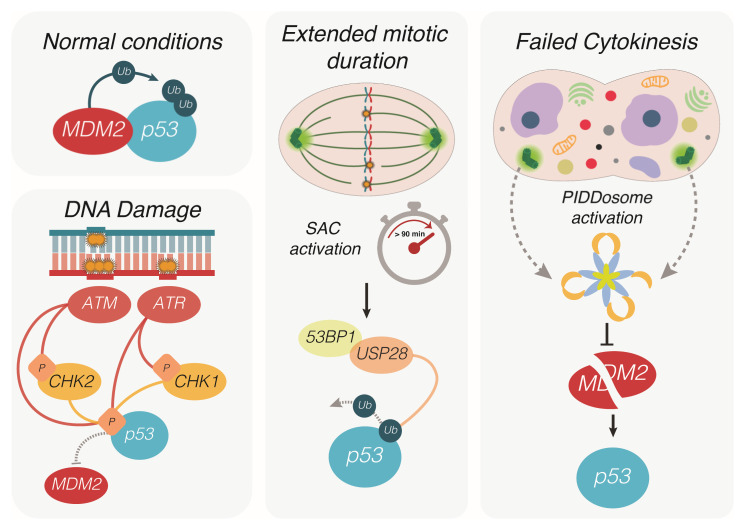
Different routes leading to p53 activation. In physiological conditions, p53 is bound and ubiquitinated by its negative regulator MDM2, which prevents nuclear translocation and promotes its proteasomal degradation. Upon single strand or double strand DNA damage, the kinases ATR and ATM activate the checkpoint kinases Chk1 and Chk2, which contribute to p53 phosphorylation on specific amino acidic residues. Phosphorylated p53 can no longer be bound and degraded by MDM2, resulting in protein stabilization, nuclear translocation and the activation of its transcriptional program. During mitosis, prolonged prometaphase due to the activation of the spindle assembly checkpoint (SAC) is sensed by 53BP1 and USP28. The latter protein promotes de-ubiquitination of p53, arresting the cell cycle in the next interphase. Defective cytokinesis prevents daughter cells to separate completely at the end of mitosis, resulting in a single polyploid cell containing extra centrosomes. The multiprotein complex PIDDosome senses the presence of extra centrosomes and functions as an activating platform for caspase-2. Being a target of this protease, MDM2 is cleaved and thereby inactivated, resulting in the accumulation of p53 and cell cycle arrest of the polyploid cell.

**Figure 2 ijms-22-10883-f002:**
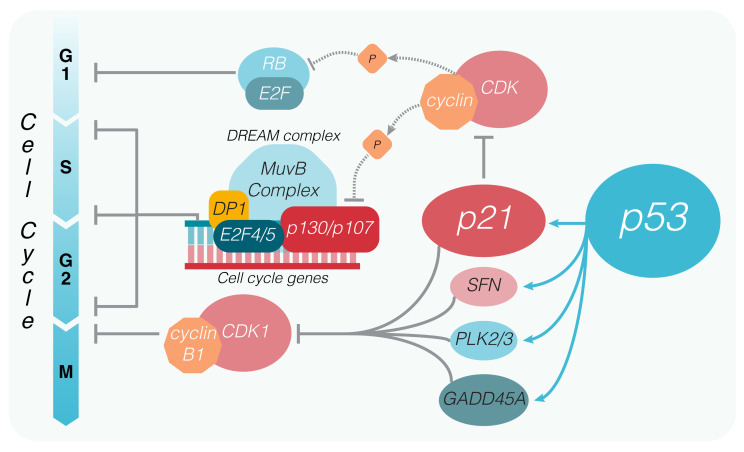
Cell cycle arrest imposed by p53. P53 controls the cell cycle via different routes by promoting the transcription of multiple targets, among which there are p21, SFN (14-3-3-σ), GADD45A and the Polo like kinases (PLK) 2 and 3. The latter proteins contribute to the inhibition of the Cyclin-dependent kinase 1 (CDK1), which upon binding to cyclin B1 promotes mitotic entry. P21 inhibits the formation of cyclinE/A-CDK2 and cyclinD-CDK4/6 complexes, which control the entry into the S phase by phosphorylating the retinoblastoma (pRB) protein. The p21-dependent inhibition of cyclin-CDK complexes also results in hyperphosphorylation of the retinoblastoma related pocket proteins p130/p107, keeping the DREAM complex in its inhibitory state, thus preventing the expression of multiple other genes responsible for cell cycle progression.

**Figure 3 ijms-22-10883-f003:**
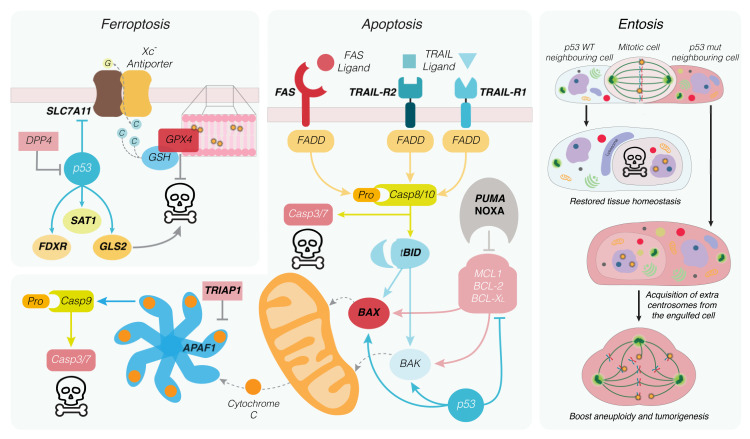
Critical p53-regulated cell death modalities. Upon alterations of the cellular redox state, iron-dependent peroxidation of membrane phospholipids can occur, promoting ferroptosis. Peroxidation can be reversed by the selenocysteine-containing protein glutathione peroxidase 4 (GPX4) in a glutathione (GSH)-dependent manner. By inhibiting the transcription of the system *Xc*^-^ complex subunit SLC7A11, p53 reduces the reservoir of cysteine (C) necessary for GSH synthesis, thus preventing reversion of lipid peroxidation. Moreover, p53 controls the transcription of FDXR, SAT1 and GLS2, favouring cell death via ferroptosis. Extrinsic apoptosis is initiated by e.g. FAS or TRAIL ligand binding to their cognate receptors promoting the multimerization of receptor trimers. These form the cell death inducing signalling complex (DISC) that recruits the adapter protein FADD, which promotes activation of the initiator caspase-8, which activates in turn executioner caspases-3 and -7, alongside the BCL2 family protein BID. Intrinsic apoptosis depends solely on the balance between the inhibitory (e.g. MCL1, BCL2, BCLX) and the apoptosis-promoting members of the BCL2 family. Regardless of the apoptotic route, BAX and BAK homo-oligomerization in the mitochondrial outer membrane results in release of cytochrome C, which complexes to APAF1 forming the apoptosome. This multiprotein complex allows dimerization of the initiator caspase-9, which in turn activates the cytosolic executioner caspases-3 and -7, leading to cell death. Cells experiencing defective mitosis are prone to form cell in cell (CiC) structures with neighbouring cells in a process called entosis. In a p53 wildtype context, the internalized cell is killed by lysosomal degradation, preventing the outgrowth of defective cells. On the contrary, entosis fosters tumorigenesis in p53 mutated cells by interfering with the mitotic process of the internalizing cell and promoting aneuploidy. Direct transcriptional targets of p53 are labelled in bold.

## Data Availability

Not applicable.

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
