# Peer review of "At a Crossroads to Cancer: How p53-Induced Cell Fate Decisions Secure Genome Integrity"

_ijms, 2021, doi:10.3390/ijms221910883_

Round 1

Reviewer 1 Report

Journal: Int J Mol Sci

Manuscript number#: ijms-1386808

Article type: Review

Title: At A Crossroads to Cancer: How P53-Induced Cell Fate Decisions Secure Genome Integrity

In this review paper, the authors discuss how mitotic errors alert the p53 network and give an overview on multiple ways how p53 can trigger cell death. They argue that a comparative analysis of different types of p53 responses, elicited by different triggers in a time-resolved manner in well-defined model systems is critical to understand cell type specific cell fate induced by p53 upon its activation, in order to re-solve the remaining mystery of its tumor suppressive function. Although this review is well described in an appropriate manner, a minor revision is required to pursue a high quality of the journal.

Minor comments

A summary figure needs to put to explain p53 roles in p53-induced cell fate decisions to secure genome integrity.

Author Response

We thank this reviewer for the overall positive evaluation. We realize now that there must have been an issue with uploading the figures. In fact, we have prepared three figures to support our written text that are now included and hopefully also accessible to this referee.

Reviewer 2 Report

Comments for the authors:

This review paper discusses how p53- induced cell fate decisions secure genomic integrity. While the topic is well studied, this review is well written but needs additional information to make it a complete review paper.

Some of the questions that need clarification are:

  1. What is the role of p53 polymorphisms and the polymorphisms associated with genes involved in p53 pathways on cancer?
  2. Are p53 activation and its ability to induce cell cycle arrest alone is sufficient for inducing senescence? If such cellular senescence is a reversible phenomenon? Is there any condition in which p53 activation serves as anti-senescence activities?
  3. P53 is a protein that is known to undergo many posttranslational modifications. What are the roles these posttranslational modifications play on cell fate decisions?

In a section on non-apoptotic cell death forms regulated by p53, the authors should include Autophagic cell death, Paraptosis, pyroptosis, and efferocytosis

Author Response

We thank this reviewer for his/her valuable feedback. We have addressed the shortcomings of our first submission by

  1. discussing the potential impact of p53 polymorphisms on page 4 of the revised version
  2. include a short note on the role of p53 in senscence and citing relevant literature on page 8, as this topic has been addressed by others extensively
  3. a discussion on the impact of different posttranslational modifications play on cell fate decisions on page 14
  4. include a short paragraph on the putative link between p53 and autophagic death or paraptosis, on page 13.

We hope that this will satisfy the reservations of this referee.

Round 2

Reviewer 2 Report

I thank you the authors for taking their time to increase the quality of their manuscript.